# Dynamic stiffening of the flagellar hook

Ashley L. Nord [1], Anaïs Biquet-Bisquert [1], Manouk Abkarian[1], Théo Pigaglio[2], Farida Seduk[2], Axel Magalon [2] & Francesco Pedaci [1]✉

For many bacteria, motility stems from one or more flagella, each rotated by the bacterial flagellar motor, a powerful rotary molecular machine. The hook, a soft polymer at the base of each flagellum, acts as a universal joint, coupling rotation between the rigid membrane-spanning rotor and rigid flagellum. In multi-flagellated species, where thrust arises from a hydrodynamically coordinated flagellar bundle, hook flexibility is crucial, as flagella rotate significantly off-axis. However, consequently, the thrust applies a significant bending moment. Therefore, the hook must simultaneously be compliant to enable bundle formation yet rigid to withstand large hydrodynamical forces. Here, via high-resolution measurements and analysis of hook fluctuations under dynamical conditions, we elucidate how it fulfills this double functionality: the hook shows a dynamic increase in bending stiffness under increasing torsional stress. Such strain-stiffening allows the system to be flexible when needed yet reduce deformation under high loads, enabling high speed motility.

[1] Centre de Biologie Structurale, Univ. Montpellier, CNRS, INSERM, Montpellier, France. [2] Aix Marseille Université, CNRS, Laboratoire de Chimie Bactérienne (UMR7283), IMM, IM2B, 13402 Marseille, France. ✉email: francesco.pedaci@cbs.cnrs.fr

Many soft biological materials, over a variety of scales, exhibit a non-linear elastic behavior wherein they become stiffer upon deformation[1]. Examples include the fibrin gels responsible for blood clotting[2], actin filaments within cytoskeletons[3], corneal tissue[4], the walls of blood vessels[5], collagen fibers of tendons[6], and the lung's extracellular matrix[7]. While such strain-stiffening often serves a critical physiological role, preventing damage upon large deformations, the molecular and structural principles underpinning the behavior remain largely unknown. An exquisite example of fine-tuning of biological mechanical properties is the rotating flagellum which provides the thrust for many motile bacteria. Rotation is supplied by the bacterial flagellar motor (BFM), a large and powerful rotary engine. Rotation of the cytoplasmic rotor is coupled to the rod, the central drive shaft, then transmitted via the hook, a short (55−60 nm) extracellular polymer, to the microns long flagellum[8,9]. The rod, hook, and flagellum are helical, hollow, slender rods, and the proteins which compose them share high sequence homology with similar quaternary structure and symmetry. Yet, while the rod is straight and rigid, the hook is supercoiled and flexible, and the flagellum is supercoiled and rigid[10–15]. Together, they provide an intriguing example of how similar sequences and structural motifs can beget strikingly different mechanical properties.

Rotating at speeds up to hundreds of Hertz and propelling the cell at tens of microns per second, the system is subject to high hydrodynamical loads, and its integrity relies upon its rigidity. But, both theoretical and experimental studies show that flagellar bundling in multi-flagellated (peritrichous) bacteria is impossible if the hook is not sufficiently flexible[16–19]. Moreover, the hook length is tightly controlled; mutations which shorten it, thereby increasing the bending stiffness, disrupt the universal joint function, whereas longer hooks lead to bundle instability and impaired motility[20]. Thus, bacterial motility relies upon a delicate combination of flexibility and rigidity in a single appendage. Previous work has shown that the relaxed hook is so flexible that it buckles under compression, an instability that monotrichous bacteria exploit to reorient their swimming direction[15]. This effect is likely more important in peritrichous bacteria, where the thrust of the off-axis flagellum applies a bending moment to the hook. But, how is the hook both soft enough to enable bundle formation yet rigid enough to withstand the force of the rotating flagellum? In monotrichous bacteria, it has been proposed that the hook becomes stiffer under increasing load[15]. However, measurements of the hook's dynamic bending rigidity under changing conditions are lacking.

Here, we use high-resolution measurements of a flagellum-tethered micro-bead and fluctuation analysis to dynamically quantify hook bending stiffness in peritrichous *Escherichia coli*. Our measurements reveal a clear dynamic stiffening of the hook as a function of motor speed, which scales with the imposed twist. Earlier measurements have provided evidence for a torsional-strain induced increase in the torsional stiffness[21,22]. Here we provide quantitative, in-vivo, and time-resolved evidence for a dynamic torsional-strain induced increase in the bending stiffness of the bacterial hook, adding this universal joint to the list of strain-stiffening biopolymers. This mechanical phenomenon may prove to be widespread among bacteria, despite different mechanisms of motility and differences in composition and sequence length of the hook protein[23].

## Results

### Radial fluctuations in bead assays. 
In the past decades, many BFM inner mechanisms have been elucidated via bead assays,

utilizing the tangential displacement of a microscopic bead tethered to and rotated by one motor (Fig. 1a, b)[24,25]. However, the microscopic description of the system still remains vague. Here we show that the radial displacement of the bead along its trajectory reports upon the mechanical properties of the hook. In particular, we observe that the bead's radial fluctuations decrease with increasing motor speed. This, via a simplified geometrical model, provides information about the bending properties of the hook.

To change motor speed, we performed resurrection experiments[26] on a non-switching strain of *E. coli* (see Methods). In Fig. 1c we show one trace displaying a clear change of radial fluctuations with increasing speed. Tracking beads in three dimensions, we observe that the bead center explores an approximately hemispherical surface while rotating, starting far from the membrane at low speed, and ending on a larger circle closer to the surface at higher speed (Fig. 1i, similar measurements are shown in Supplementary Fig. 1 and Supplementary Fig. 2). We summarize these observations in Fig. 1j, which sketches the geometry we define in more detail below.

**Geometrical model.** In Fig. 2a we outline a plausible simplified geometry of the bead tethered to the filament stub, composed by the hook and the flagellar stub (protein FliC). The tangential variable commonly tracked in bead assays is the angle $\phi$, together with its time derivative $\omega$. The bead (of radius $R_b$), once hindered by the surface (considered planar for simplicity), cannot spin around the tilted flagellar axis but can only rotate around the vertical motor axis $z$. With smaller beads, or longer tethers, this constraint is relaxed[27]. The radius $r$ indicates the distance of the bead center from the center of the $x$, $y$ trajectory (examples of $r(t)$ can be seen in Fig. 1d–g). In the $(z, r)$ plane (Fig. 2b), $L$ is the distance between the bead center and the origin (where the hook intersects the membrane) and $\theta$ is the angle between $L$ and the membrane. We assume that the hook is an angular spring[13,15], and that $L$ is constant due to the negligible extension of the hook and filament, so changes of $\theta$ reflect the bending of the hook (as also suggested in ref. [28]). The gap $s$ between the bead and the membrane, changing with $\theta(t)$ and $r(t)$, is not directly measured, as the bead position is tracked relative to an arbitrary origin. To quantify $s(t)$, we determine its minimum value $s_{\min}$ over the entire trajectory following the analysis described below. To summarize our assumptions (Fig. 2a–c): at a given time $t$, the bead $x(t)$, $y(t)$, $z(t)$ position is at a constant distance $L$ from the motor, described by the radius $r(t)$ and angle $\theta(t)$, corresponding to a bead-membrane gap $s(t)$. Our analysis focuses in particular on the fluctuations of the angle $\theta(t)$.

**Drag coefficients.** In this overdamped system, the proximity to the membrane must be taken into account to correctly quantify the bead viscous drag. An accurate value of the drag is required to determine the motor torque. For a displacement parallel to the $(x, y)$ plane in the direction of the tangential linear speed $v$ (Fig. 2a), we calculate the drag $\gamma_\phi$ (function of $s$, $R_b$, $\langle r \rangle$) employing both Brenner and Faxen theory, depending on the ratio $s/R_b$, to correct for the vicinity of the surface[29–31], as detailed in Supplementary Methods. The motor torque can then be calculated by $\tau = \gamma_\phi \omega$. For a displacement in the $(r, z)$ plane, the drag has components parallel and perpendicular to the membrane, $\gamma_\parallel = \gamma_o C_\parallel(s, R_b)$ and $\gamma_\perp = \gamma_o C_\perp(s, R_b)$, respectively (Fig. 2c), where $\gamma_o$ is the bulk drag and the correction terms $C_\perp$, $C_\parallel$ are defined in Supplementary Methods.

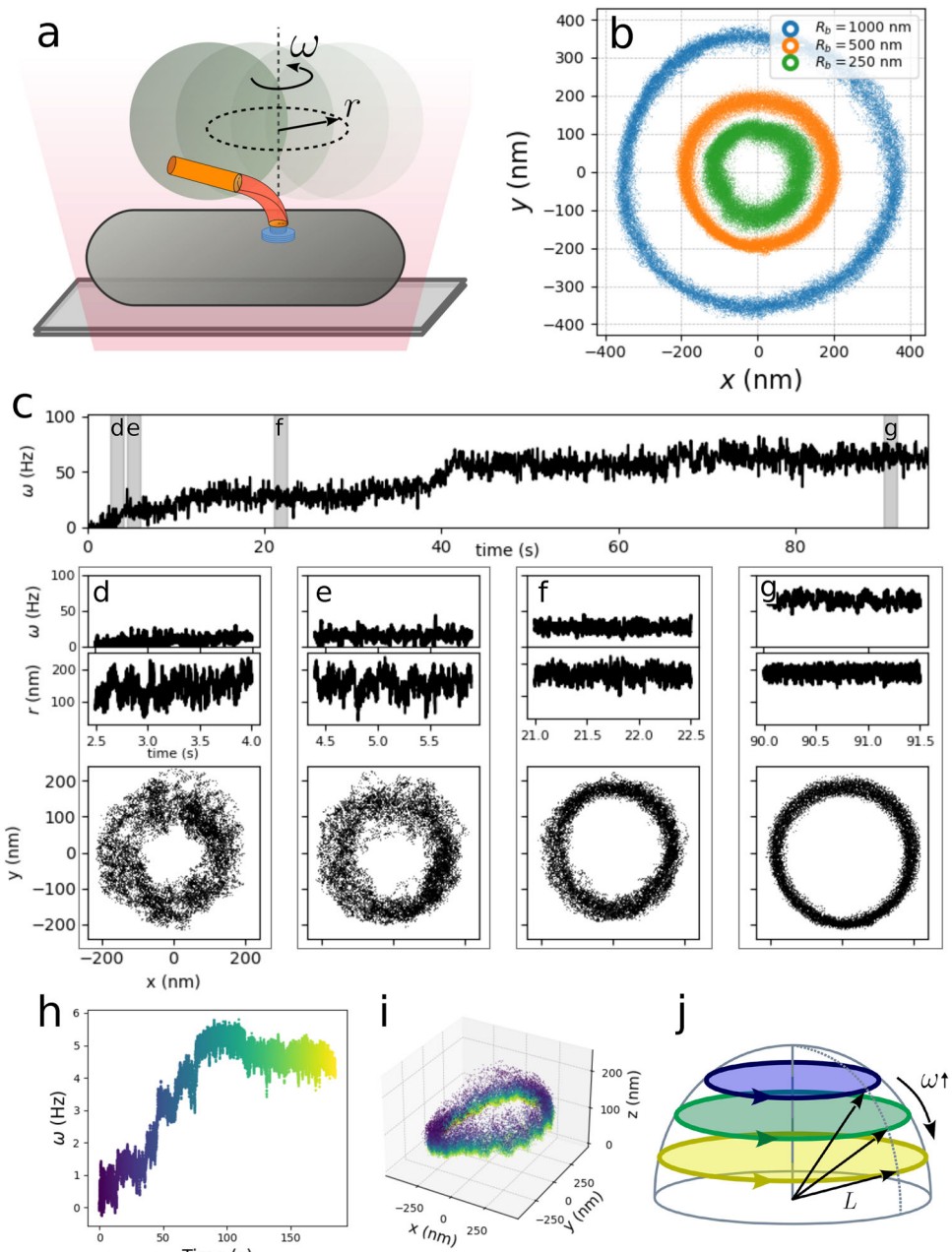

**Fig. 1 Bacterial Flagellar Motor bead assays. a** Schematic experimental setup (not to scale). A living bacterium is adhered to the microscope glass slide, and one microscopic bead is attached to the filament stub, rotating on an approximately circular trajectory, observed by optical microscopy and tracked at 8–10 kHz sampling rate. **b** Experimental x, y trajectories of the bead center obtained for beads of different diameter. **c** Resurrection trace ($R_b = 500$ nm, to increase readability the trace is here filtered with a 8 ms running window median filter). Along the time trace of the motor speed $\omega(t)$, four time-windows are highlighted, and for each window the corresponding signals $\omega(t)$, radius $r(t)$, and the bead trajectory $(x(t), y(t))$ are shown in the respective **d**–**g**. The decrease in radial fluctuations at increasing speed is particularly evident. **h** A resurrection trace (from a different motor than in **c**), where the color code indicates the time. **i** 3D tracked position of the bead corresponding to the measurement in **h**), with the same color code for time. The bead starts far from the membrane for low $\omega$ and approaches it when rotating faster, tracing circles of larger diameter. The trajectory is tilted because of the presence of the cell body. **j** Schematic representation of the behavior of the trajectory in **i**. We simplify the geometry assuming that the bead center moves on circular trajectories on a hemispherical surface of fixed radius $L$. The color code indicates time as in **h** and **i**.

As we focus on the movement of the bead along $\theta$, the corresponding drag is

$$\gamma_\theta = L^2(\gamma_\parallel \sin^2\theta + \gamma_\perp \cos^2\theta). \qquad (1)$$

We note that for each measurement, all the parameters discussed above and defined in Fig. 2, can be quantified from the measured $x(t)$, $y(t)$ position of the bead, after determination of the minimum gap $s_{\min}$.

**Fluctuation analysis.** The radius $r(t)$ and the angle $\theta(t)$ describe the motion of a point thermally fluctuating in a potential well in a rotating reference frame. By analyzing the fluctuations of the signals, the shape of the potential as a function of time can be characterized, yielding information on the elastic properties of the physical tether. In order to characterize the fluctuations in $\theta(t)$, we divide each experimental trace in time-windows (Fig. 3a), and in each window $i$ we calculate the mean and variance of the signal

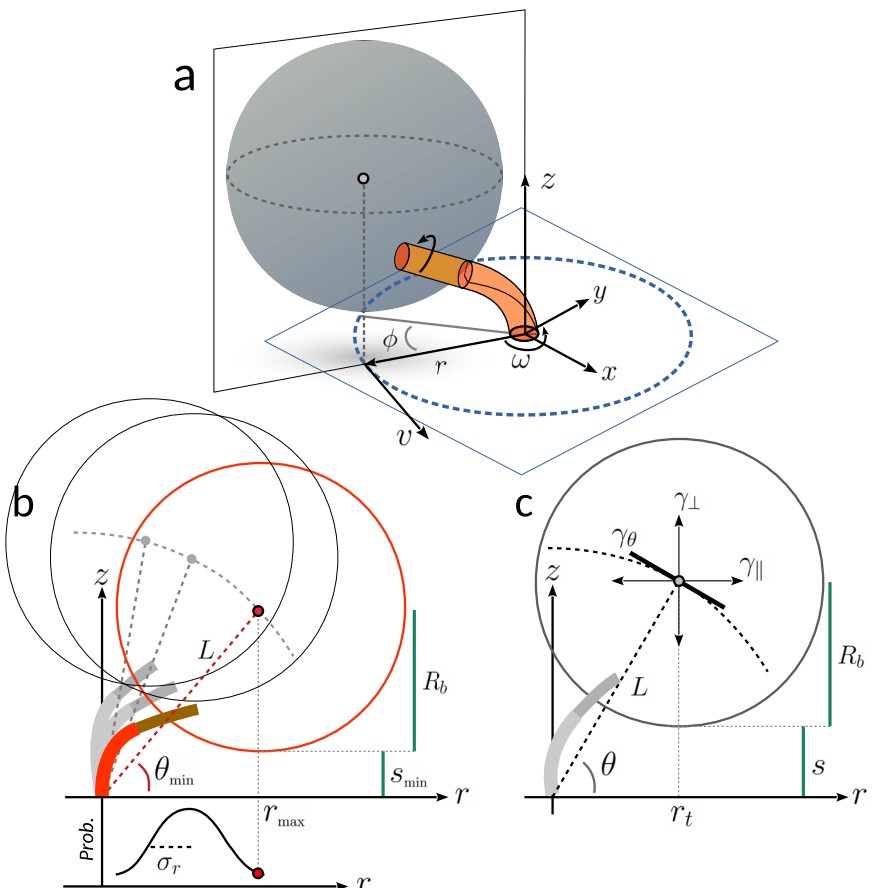

**Fig. 2 Microscopic geometrical model. a** 3D representation of the bead (not to scale) tethered to the flagellar stub, composed by the hook (red, length of 60 nm) and filament (FliC$^{st}$, orange, of 20-nm diameter[32]). The angle $\phi$ describes the motion of the center of the bead along the circular trajectory of radius $r$. The linear speed of the bead is $v = \omega r$, where the angular velocity is $\omega = d\phi/dt$. **b** Projection on the ($z, r$) plane, where the center of the bead is assumed to move on an arc of radius $L$, described by the angle $\theta$. The probability distribution of $r$ (approximated by a Gaussian of width $\sigma_r$) is shown in the graph below. The minimum value of $\theta$ visited in an entire measurement ($\theta_{min}$) corresponds to the maximum visited value of the radius $r_{max}$, and to the minimum distance $s_{min}$ between the bead surface and the membrane. **c** Same as in **b** for a generic position ($r_t, \theta$) at time $t$. The drag coefficient on the plane ($z, r$) is composed by a parallel ($\gamma_{\parallel}$) and perpendicular ($\gamma_{\perp}$) component with respect to the membrane, which are projected ($\gamma_{\theta}$) on the direction tangent to the arc of radius $L$. The distance between the bead surface and the membrane is $s(t) \geq s_{min}$. The radius of the bead is $R_b$.

$\theta_i(t)$ from its probability distribution, which is well fit by a Gaussian function (Fig. 3c1–c3), indicating a harmonic potential whose stiffness $\kappa_{\theta}$ can be quantified by the Equipartition theorem. We also calculate the power spectral density of $\theta_i(t)$, $PSD_{\theta_i}(f)$, a function of frequency $f$ (Fig. 3d1–d3), which provides information on both the stiffness and drag coefficient. Borrowing from the analysis employed for objects fluctuating in optical harmonic potentials[33], we fit the experimental $PSD_{\theta_i}(f)$ with the Lorentzian function $L(f) = (k_BT)/(\pi^2\gamma_{\theta}(f^2 + f_c^2))$, where $k_BT$ is the thermal energy (Fig. 3d1). The fit yields the drag $\gamma_{\theta}$ defined above (Fig. 3d3) and the corner frequency $f_c$ (Fig. 3d2), related to the potential stiffness by $\kappa_{\theta} = 2\pi\gamma_{\theta}f_c$. The duration of the time-windows was chosen to sufficiently sample the plateau of $PSD_{\theta_i}(f)$ at low frequency, while the high sampling rate of the camera allows to sample frequencies higher than $f_c$ (Fig. 3d1). As the example in Fig. 3c1–c3 shows, for an increasing motor speed $\omega$, the distribution of $\theta$ tends to become sharper and closer, in average, to the membrane (i.e. both $\sigma_{\theta}$ and $\langle\theta\rangle$ decrease), and better approximated by a Gaussian. At the same time, both the corner frequency $f_c$ and the drag $\gamma_{\theta}$ increase (Fig. 3d2, d3). This reflects the bead being pushed towards the membrane for increasing $\omega$, where the drag becomes higher because of surface proximity, while the elastic tether becomes stiffer in bending. In

Supplementary Note 1, we use Langevin simulations to show that these observations, and in particular the increase in the measured corner frequency and stiffness, cannot be explained solely by the hydrodynamic increase in drag due to the wall proximity. We also show (Supplementary Note 4) that the centrifugal force cannot explain the observed bead displacement. Thus, our measurements point to an increasing bending stiffness of the hook.

Not being directly measurable, we estimate the gap $s_{min}$ by minimizing the mean square error between the theoretical value of the drag $\gamma_{\theta}$ of eq. (1) (controlled by $s_{min}$) and the experimental value obtained from the Lorentzian fit of the spectrum. The full procedure, described in detail in Supplementary Methods, allows us to determine the values of all the quantities defined above. In Supplementary Methods, we also describe how the analysis is modified for the traces where we also measure $z(t)$, where the assumption of a constant $L$ (necessary only when $x, y$ are detected) is justified by the data.

In Fig. 4 we plot the probability distributions for $\langle r \rangle$, $s_{min}$, and the two drag coefficients $\gamma_{\phi}$ and $\gamma_{\theta}$, resulting from the analysis performed on all the measured traces for three bead sizes. For smaller bead radius $R_b$, the average radius $\langle r \rangle$ of the trajectory decreases (Fig. 4a and Fig. 1b), as well as the value of the minimum distance $s_{min}$ (Fig. 4b). When scaled by the bead radius, the three distributions of $s_{min}$ are similar (Fig. 4b, inset),

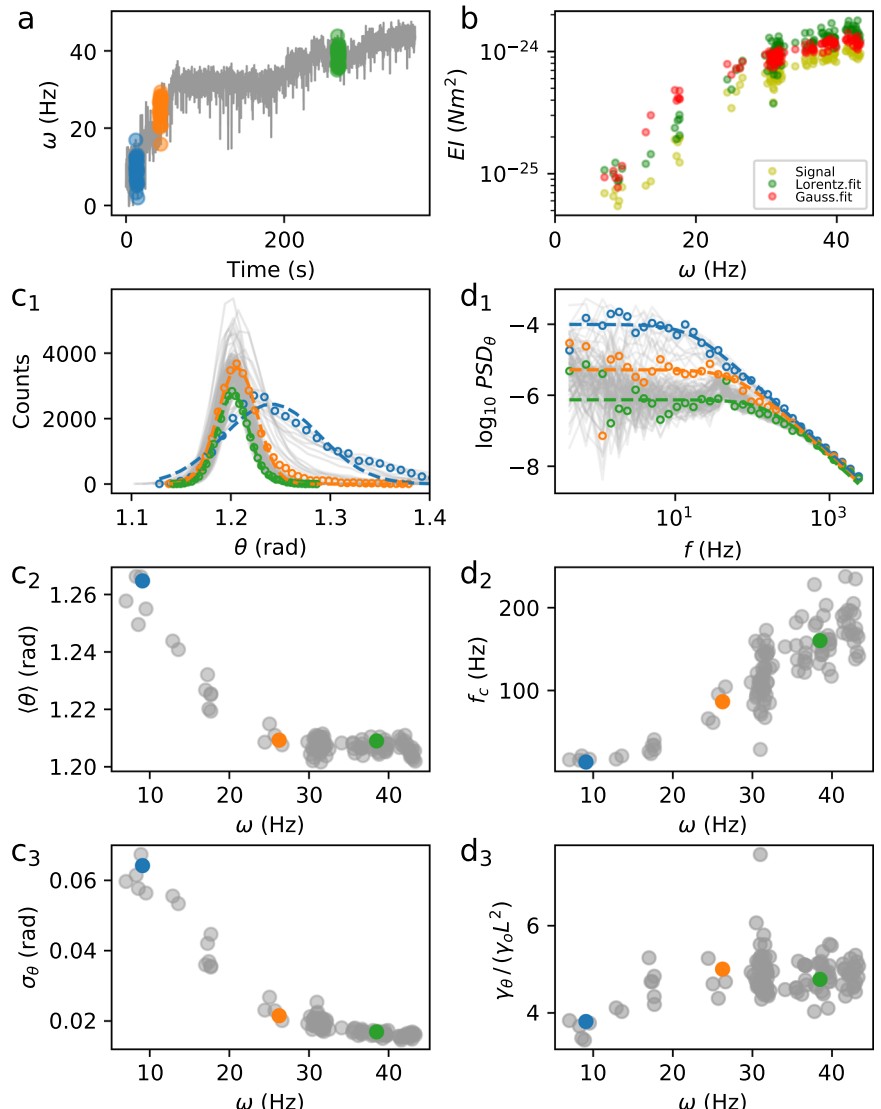

**Fig. 3 Single motor fluctuation analysis. a** One resurrection trace of the speed $\omega$ ($R_b = 500$ nm, to increase readability the trace is here filtered with a 8 ms running window median filter) is divided in time-windows of 3$s$ to allow local fluctuation analysis of the signal $\theta(t)$. Three windows are highlighted, and their colors are used for the corresponding points in all the panels. **b** Bending stiffness $EI$ as a function of speed $\omega$, calculated in each time-window following the analysis described in the text, using the raw signal $\theta_i(t)$, the Gaussian fit to its probability distribution, and the Lorentzian fit to its power spectral density. **c1** Probability density of all the $\theta_i$ along the trace (gray lines, while the points and their colors correspond to the windows shown in **a**), fit by a Gaussian function (dashed lines for the three windows highlighted). **c2** Mean value $\langle\theta_i\rangle$ as a function of the mean speed $\omega$ in the time-window. **c3** Standard deviation of $\theta_i$ as a function of the mean speed $\omega$ in the time-window. **d1** Power spectra $PSD_{\theta_i}(f)$ (gray lines) of all the time-windows along the trace, and Lorentzian fit (dashed lines) for the three windows highlighted in **a** (points). **d2** Corner frequency $f_c$ of the Lorentzian fit in **d1** as a function of speed $\omega$. **d3** Drag coefficient $\gamma_\theta$ from the Lorentzian fit, normalized to the bulk value $\gamma_o L^2$.

indicating that the bead minimum distance from the membrane is of the order of $0.1R_b$. Finally, the measured value of the drag $\gamma_\phi$ (Fig. 4c), calculated using $s_{\min}$, is used to calculate the motor torque ($\tau = \gamma_\phi \omega$, see Fig. 5). The experimental values of the drag $\gamma_\theta$ in the plane ($r$, $z$) (Fig. 4d) result from the procedure used to determine $s_{\min}$, and therefore are in agreement with the theoretical values obtained by eq. (1).

**Hook bending stiffness**. The hook is the most flexible part of the tether which can bend and twist. While the angle $\theta$ does not correspond to the real bending angle of the hook, its variations are the same. Therefore, the bending stiffness of the hook can now be determined from the fluctuations of $\theta_i(t)$ in each time-window $i$ of the trace. Assuming a length of the hook $L_{\text{hook}} = 60$ nm[15], the bending stiffness $EI$ (where $E$ is Young's modulus and $I$ is the area

moment of inertia) can be quantified using the Equipartition theorem by $EI = k_B T L_{\text{hook}}/\sigma_{\theta_i}^2$, where the variance $\sigma_{\theta_i}^2$ can be found either from the raw $\theta_i$ signal or from its Gaussian fit. Alternatively, the bending stiffness can be determined in each time-window from the values of $\gamma_\theta$ and $f_c$ provided by the Lorentzian fit of the spectrum, by $EI = 2\pi\gamma_\theta f_c L_{\text{hook}}$. The three calculations (different but not fully independent) lead to similar results, as shown in Fig. 3b for a single trace, where $EI$ increases by one order of magnitude from $10^{-25}$ to $10^{-24}$ Nm$^2$, as the motor speed increases. Resolving with high resolution the speed changes due to stator incorporation helps us to quantify $EI$ over a range of speed and torque values.

At steady rotation, the torque produced by the motor is stored as twist of the hook. Such a torsional elastic link has a low-pass filtering effect on the measured position and speed of the bead, and is particularly problematic for the resolution of the fundamental

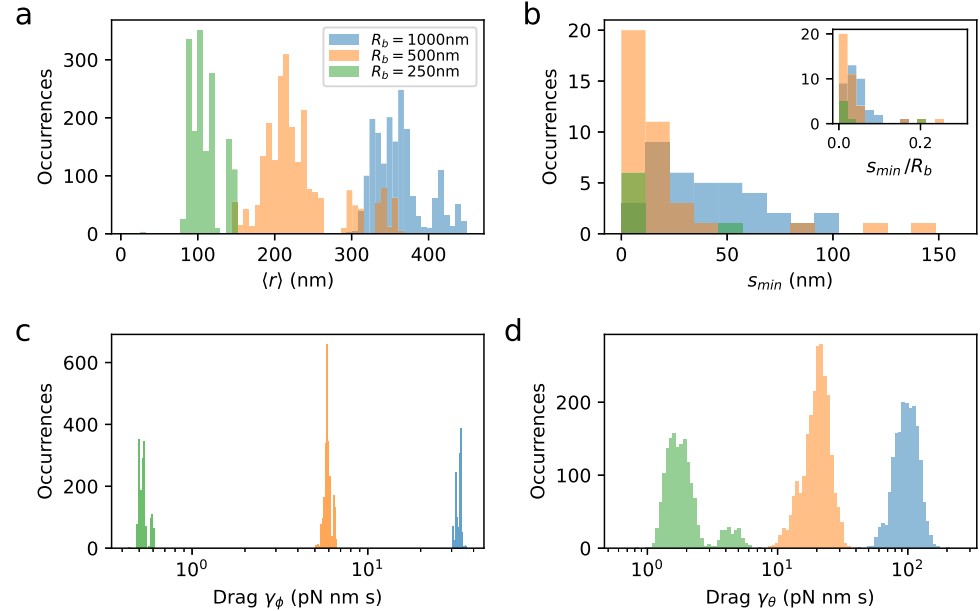

**Fig. 4 Experimental distributions of parameters defined in Fig. 2, obtained from the analysis of all the measured traces for the three loads considered.**
**a** Histograms of the average radius $r_i$ in each time-window of the $x$, $y$ bead trajectories. **b** Histogram of the optimal value of the distance between the bead and the membrane $s_{min}$, where one value is obtained from the analysis of each trace. Inset: distributions of $s_{min}$ normalized by the value of the bead radius $R_b$. **c**, **d** Distributions of the drag coefficients $\gamma_\phi$ and $\gamma_\theta$, obtained in each time-window of all the traces. The color code, indicating the bead radius, is the same for all the panels. The measurements, each on a different cell, consist of 36, 35, and 7 traces for motors at steady state and 6, 13, and 6 resurrection traces for beads of radius $R_b = 1000$, 500, 250 nm, respectively.

step of the motor[34,35]. In Fig. 5 we characterize the hooks of traces acquired for the different loads as a function of motor torque. Using the published value of the hook torsional stiffness ($k_\phi = 400$ pN nm/rad[21,22]) as a first approximation, we can convert motor torque to twist, where the maximum torque of 2000 pN nm corresponds to a maximum twist of 280° (likely overestimated, as this is beyond the linear elastic region of ~180–270° reported in ref. [21,22]). Figure 5a shows the individual trajectories of each hook in the plane ($\tau$ or twist, $EI$), where the torque produced in each time-window of the traces is calculated by $\tau = \gamma_\phi \omega$, where $\omega$ is the angular speed of the bead and $\gamma_\phi$ is the drag, which includes the rotation and translation of the bead as well as as the correction due to the proximity to the membrane (see sec."Drag coefficients" above and Supplementary Methods). Despite the heterogeneity, it is clear that when the torque of the motor, and therefore the twist of the hook, increases due to a sufficiently high load, the stiffness $EI$ of an individual hook can increase by a factor 10−20. Figure 5b shows in the same plane all the experimental points considered (one per time-window of each trace), used to build the trajectories in Fig. 5a. We note that the trajectories of all the different loads globally converge at low torque in the region $EI \sim 0.5 - 1 \times 10^{-25}$ Nm$^2$, giving an estimate of the bending stiffness of the torsionally relaxed *E.coli* hook. This range is compatible with the measurement in *V. alginolyticus*[15]. A linear function (Fig. 5b, dashed line) fits reasonably well the binned average points (dark) of the plane. In Supplementary Note 3 we consider the effects of speed fluctuations combined with the observed decrease of $\langle\theta\rangle$ with speed, concluding that the effect of speed fluctuations cannot obviously explain the observed trend of $EI$ increasing with speed.

In Fig. 5c we show that our analysis can further characterize the stiffness $EI$ both as a function of twist and bending angle, reported by torque $\tau$ and angle $\langle\theta\rangle$ (averaged in each time-window), respectively. The experimental points populate the 3-dimensional space ($EI$, $\langle\theta\rangle$, $\tau$), where the background colors indicate the average torque in each region. The stiffening observed in Fig. 5a, b is resolved here for different values of

bending angle $\langle\theta\rangle$. Moving vertically in the plot at constant $\langle\theta\rangle$, $EI$ increases with increasing twist and torque (reported by the color of the points). On the other hand, moving horizontally, an increasing bending angle at constant twist (color) is not accompanied by an appreciable change in stiffness $EI$. In other words, the bending stiffness increases with twist but not appreciably with bending angle, in the explored range.

The above results show that the hook stiffens when twisted by a motor rotating counter clockwise (CCW). Given the complex coiled structure of the hook[13,14], we asked whether the hook would respond asymmetrically to twist applied in the opposite direction. To compare the bending stiffness of a single hook twisted in both directions, we employed a mutant strain ($\Delta cheRB$) which increases the fraction of time the motor spends rotating CW (see Supplementary Methods). The results, shown in Supplementary Note 5, indicate that in a majority of cases the bending stiffness under CCW twist was higher than under CW twist. However, the heterogeneity of these results is large and further investigations are required to give a definitive answer and resolve a possible asymmetry.

## Discussion

Our high-resolution data and fluctuation analysis reveal the hook to be a strain-stiffening polymer, showing a linear increase in hook bending stiffness, by more than one order of magnitude, as a function of the torque (and therefore twist) produced by the motor. The hook persistence length $L_p$ can be calculated from the bending stiffness $EI$ and the thermal energy $k_BT$, as $L_p = EI/k_BT$. Our measurements show (in agreement with those of ref. [15]) that the stiffness increases in the range $EI \sim 5 \times 10^{-26} - 3 \times 10^{-24}$ Nm$^2$, yielding a persistence length of about 10 μm in the hook's relaxed state, and up to several hundreds of microns in the torsionally loaded state. From the area moment of inertia $I$ of a hollow cylinder, we can also estimate the Young's modulus $E$ of the hook, found in the range $10^7 - 10^9$ Pa (see Supplementary Methods). Considering other tubular biopolymers, this sets the hook between microtubules ($L_p > 1$ mm) and F-actin ($L_p \sim 10$ μm)[36]. It is worth

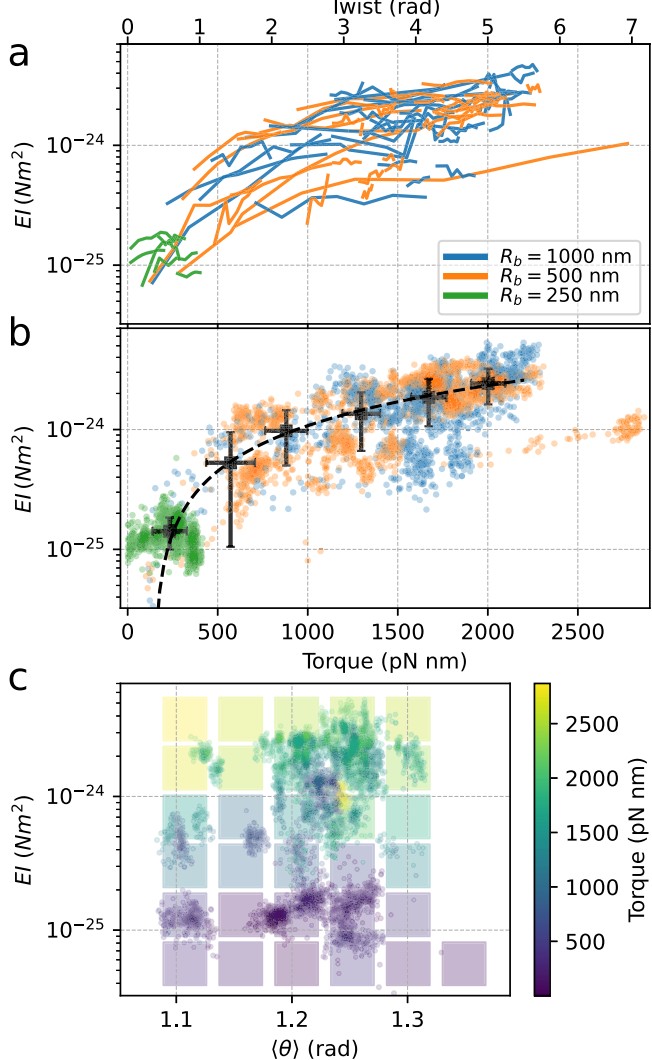

**Fig. 5 Hook bending stiffness. a, b** Bending stiffness as a function of motor torque $\tau$ (bottom axis) and twist angle (top axis, calculated from a constant value of the torsional stiffness), for the three loads considered. **a** the lines indicate the trajectories followed by individual motors during resurrection. **b** all the experimental points (one for each time-window) are shown, and are binned together in six points (black points represent the mean ± SD in each bin). The dashed line is a linear fit to the black points. **c** The experimental points are shown in the space $(EI, \langle\theta\rangle, \tau)$, where $\langle\theta\rangle$ is a proxy for the bending angle, and the torque $\tau$ for the twist angle. The background colors indicate the average torque value in the corresponding region of the plane $(EI, \langle\theta\rangle)$. The measurements, each on a different cell, consist of 36, 35, and 7 traces for motors at steady state and 6, 13, and 6 resurrection traces for beads of radius $R_b = 1000$, 500, 250 nm, respectively.

different repeat distance, which varies periodically with motor rotation. An atomic model of FlgE from *Salmonella* has shown 11 distinct conformations of FlgE, with the subunits of a given protofilament having the same conformation. A superposition of the 11 subunits shows a change in the relative domain orientations[13,14]. The inside bend of the hook is successively occupied by different protofilaments, and the compression and extension of each protofilament with each revolution arises due to dynamic changes in the relative orientations of the three FlgE domains about two hinge regions. This, in turn, begets dynamic changes in inter-subunit interactions, which confer upon the hook its superhelical form and bending flexibility. Hook curvature is produced by close packing interactions between D2 domains along the 6-start helix on the inner side of the bend[37,39]. The hook undergoes polymorphic transformations in response to changes in the temperature, salt concentration, or pH[40], and it is thought that the superhelical curvature and twist depend on the direction of these D2-D2 interactions. The D2 domain of subunit 0 also interacts with the D1 domain of subunit 11, and MD simulations have shown large axial sliding upon compression and extension[13,14,37]. An intrinsically disordered region that connects D0 and D1 governs inter-subunit interactions and has been shown to play a role in the stability of the hook structure[41–43]. It is therefore conceivable that, under the strain imposed by a global twist, changes in these interactions could give rise to decreased bending flexibility, a hypothesis that could be explored by future MD simulations.

The fact that stiffening in bending occurs with increasing twist suggests that a coupling may exist between these two classically decoupled directions, as described in actin filaments and DNA[41,44,45]. However, we note that this formalism does not explain a change in stiffness (see Supplementary Note 2). Twist-bend coupling produces an equilibrium bending angle that is directly modified by the twist in the structure, and therefore could be responsible for the observed approach of the bead to the surface. However, such behavior of the bead could also be explained by the fact that, during rotation, the bead would tend to rotate around the axis of the tilted flagellar stub, and eventually be pushed against the membrane.

Our results are in agreement with the first observations of hook bend stiffening in the polar *V. alginolyticus*[15] (Supplementary Table 1), where the measurements were performed on different hooks either torsionally relaxed or subject to physiological swimming-induced twist. These experiments, in combination with mathematical models[46,47], suggest that a dynamic *EI* provokes a tuned buckling of the hook, or flicking, allowing this monotrichous bacterium to change swimming direction. Yet, here we show that a dynamic bend stiffening also occurs in multi-flagellated *E. coli*. While the hook of *V. alginolyticus* appears straight[48,49], that of *S. enterica* is supercoiled[14]. Thus, the angle between the hook and membrane is natively acute, and our experiments suggest that this angle decreases slightly with twist. One may imagine that an increasingly rigid hook can produce opposite effects on the stability of the flagellar bundle in peritrichous bacteria. On one hand, a more rigid hook can withstand the bending moment coming from a tilted flagellum rotating in the bundle, increasing bundling stability. On the other hand, the universal joint function of the hook could conceivably be negatively affected by increasing bending rigidity. Our results indicate that the stabilizing effect prevails, constraining the mechanical model of a stiffening hook. Thus, bundle formation and tumbling in multi-flagellated bacteria could benefit from the same mechanism as flicking in polar-flagellated bacteria, and dynamic stiffening might be a common strategy in motile bacteria. We expect that novel single-molecule force and torque manipulation assays will provide further insights into the mechanics of this striking biopolymer.

noting that the long lever arm of the flagellum is relevant: for example, a 1 pN force applied along the flagellum 1 μm from the relaxed hook would produce a bending of 70 degrees, allowing bundle formation. The tight regulation of the physical length of the hook observed in bacteria[8,20] is likely in line with the need for careful tuning of its elastic properties.

What mechanism can explain this dynamic torsion-induced stiffening? One explanation may be a torsion-induced global restructuring of the hook, affecting its mechanical properties. The hook protein, FlgE, has three domains, and the 11 protofilaments of the hook form a short segment of a superhelix[13,14,37,38]; thus, each protofilament adopts a different length and its subunits a

## Methods

**Bacterial strains and growth**. The *E. coli* strain used was MT03 (parent strain: RP437, Δ*pilA*, Δ*cheY*, *fliC*$^{st}$), a non-switching strain in which chromosomal replacement of the wild-type flagellin gene *fliC* with the *fliC*$^{st}$ variant exposes the hydrophobic core of the flagellar filament, rendering it 'sticky' to hydrophobic surfaces such as polystyrene[50]. For experiments which investigated *EI* as a function of rotation direction, we used a strain in which we deleted *cheR* and *cheB* from MT02 (parent strain: RP437, Δ*pilA*, *fliC::Tn10*; see Supplementary Note 5 and Supplementary Table 2 for details). Bacteria cultures were seeded from frozen aliquots (grown to saturation and stored in 25% glycerol at −80°C) and grown in tryptone broth for 5 hours at 30°C, shaking at 200 rpm, until an OD$_{600}$ of 0.5–0.8. The flagellar filaments were sheared by passing the culture through a 21 G needle 50 times with a syringe. The culture was then washed and resuspended in motility buffer (MB, 10 mM potassium phosphate, 0.1 mM EDTA, 10 mM lactic acid, pH 7.0). To vary the hydrodynamic load, affecting the speed of the BFM, we employ polystyrene beads (Sigma-Aldrich) of diameter 2000, 1000, and 500 nm.

**Experimental measurements**. Custom microfluidic slides consisted of two coverslips (Menzel-Gläser #1.5) sealed by melted parafilm. The top coverslip had two holes for fluid exchange. Poly-l-lysine (Sigma-Aldrich) was introduced to the microfluidic slide, left to incubate for 5 min, then washed out with MB. Cells and then beads were sequentially introduced and allowed to sediment for 10 min, and the remnants were washed away with MB. Experiments were performed at 22 °C. Rotating beads were imaged with a custom inverted in-line holographic microscope setup[51]. The sample was illuminated by a 660 nm laser diode (Onset Electro-Optics; HL6545MG) and imaged via a 100 × 1.45 NA objective (Nikon) onto a CMOS camera (Optronics CL600x2/M) at 8.8 or 10 kHz. The *x, y* position of the bead (vectors parallel to the coverslip) were determined by cross correlation with a synthetic bead hologram. The *z* position of the bead (orthogonal to the coverslip) was determined by comparing the radial profile of the bead to profiles previously acquired at a known *z* via piezo controlled movement of the objective[52]. The bead trajectory is corrected to remove deterministic features and artifacts (Supplementary Methods). Resurrection experiments were performed by introducing carbonyl cyanide *m*-chlorophenyl hydrazone (CCCP, Sigma-Aldrich) into the microfluidic slide for 5 min, then washing it out with MB. Data acquisition and bead tracking were performed with custom Labview scripts.

**Data analysis**. Each experimental trace was divided in time-windows of 1, 3, and 4 s for beads of radius $R_b = 250$, 500, and 1000 nm, respectively. In each window, the angle $\theta(t)$ (see Fig. 2) was calculated from the $(x(t), y(t))$ or $(x(t), y(t), z(t))$ position of the bead by fitting the bead trajectory to an ellipse (see Supplementary Methods). The angular fluctuations of the hook were fit with a Lorentzian function to extract the bending stiffness of the hook, *EI*. Full details of the analysis workflow are given in Supplementary Methods. All analysis was performed with custom Python scripts.

**Reporting summary**. Further information on research design is available in the Nature Research Reporting Summary linked to this article.

## Data availability

The analyzed data generated in this study are provided in the Source Data file. Example raw data can be found in the group's code repository[53]. The raw datasets generated and analyzed during the current study are available from the corresponding author upon reasonable request. Source data are provided with this paper.

## Code availability

The code used for the analysis, as well as example data can be found at https://github.com/SMADynamics/BFM_radial_fluct https://doi.org/10.5281/zenodo.6405461[53].

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

## Acknowledgements
We thank Martin Rieu, Richard Berry, Nils-Ole Walliser, Luca Costa, and Marcelo Nollmann for fruitful discussions. The bacterial strains used in this work were gifts from the labs of Judy Armitage and Richard Berry. A.B.B. and T.P. were supported by the ANR FlagMotor project grant ANR-18-CE30-0008 of the French *Agence Nationale de la Recherche*. The C.B.S. is a member of the France-BioImaging (FBI) and the French Infrastructure for Integrated Structural Biology (FRISBI), two national infrastructures supported by the French National Research Agency (ANR-10-INBS-04-01 and ANR-10-INBS-05, respectively).

## Author contributions
A.L.N. and F.P. conceived the experiments and analysis. T.P., F.S., and A.M. constructed and characterized the mutant strains. A.L.N. and A.B.-B. performed the experiments. A.L.N., A.B.-B., M.A., and F.P. designed the analysis, and F.P. developed the analysis code. A.L.N. and F.P. wrote the first draft of the manuscript. All authors participated in data interpretation and final manuscript preparation.

## Competing interests
The authors declare no competing interests.
