## [Peer Review File · Nature Communications]

Dynamic stiffening of the flagellar hookREVIEWER COMMENTS

Reviewer #1 (Remarks to the Author):

Bacteria move in liquid environments by rotating a sophisticated multi-component nanomachine, the flagellum. The flagellum consists of three main parts: the cell-enveloped embedded basal body, the external hook and filament. The basal body includes motor-force generators that use energy derived from the cell's ion gradient across the inner membrane to rotate the cytoplasmic part of the basal body, which then transmits the rotational energy to a drive-shaft (the rod), via a flexible linking structure (the hook) to a rigid propeller (the filament). The number and placement of flagella vary widely between different bacterial species e.g. peritrichously flagellated bacteria assemble several flagella randomly across their cell surface. Of particular interest in this respect is the flexible hook structure, which is thought to function as an universal joint enabling the formation of a filament bundle independent on the off-axis placement of peritrichous flagella. Another unique feature of the hook structure is its length control mechanism, which limits its length to around 55 nm in *E. coli* and *Salmonella*. It has been shown previously that the hook's flexibility and length is crucial for proper filament bundle formation and thus bacterial motility. However, the hook needs to be able to rigid enough to withstand the force of the filament rotating at speeds of up to hundreds of Hz.

Here, Nord et al. tracked filament-tethered micro beads to quantify hook bending stiffness of the *E. coli* flagellum in order to investigate the hook's dynamic mechanical response under changing conditions. They found that the hook functions as a strain-stiffening polymer, which shows a linear increase in hook-bending stiffness as a function of motor torque.

This is a nicely written manuscript with beautifully illustrated figures that greatly enhances our mechanistic understanding of one of nature's most fascinating molecular machines, the flagellum. I only have a few comments that might help the authors to improve the clarity of their manuscript:

- 1) The number of analyzed motors should be indicated in the figure legends, as well as if the presented data are from multiple motors of a single cell or from motors of different cells.
- 2) The legends of Fig. 2 and Fig. S4 are missing/cut short?
- 3) As stated above, the length of the hook structure is rather tightly controlled (e.g. 55 nm +/- 6 nm in *Salmonella* (Hirano, T., Yamaguchi, S., Oosawa, K., and Aizawa, S. (1994). Roles of FliK and FlhB in determination of flagellar hook length in *Salmonella typhimurium*. *J Bacteriol* 176, 5439-5449)? It is still unclear, why bacteria evolved such an intricate length control mechanism of the hook. The discussion could benefit from discussing the importance of the length control for the hook bending stiffness.
- 4) Why does the *fliC::Tn10* mutation result in a "sticky" filament?
- 5) Does the *cheRB::cat* insertion have polar effects on *cheYZ* expression? In this respect, the rationale of using a *cheRB* mutant (resulting in motors that switch between CCW and CW rotational direction) instead of a *cheZB* mutant (which would allow to analyze CW-only rotating motors) is not clear.

Reviewer #2 (Remarks to the Author):

The authors study the hook that joins the rotor and flagellum in peritrichous *E. coli*. They ask how the hook can allow for bundle formation while also being able to withstand the force of the rotating flagellum. They propose that this may be achieved by an increase in the bending stiffness of the hook under increasing torsional stress.

To check their hypothesis the authors present measurements of the motion of a flagellum-tethered micro-bead at different motor speeds. These show that for increasing motor speed, the peak of the hook bending angle distribution becomes narrower and closer to the membrane. The authors estimate the bending stiffness of the hook from the bending angle distribution and find that it increases from $10e-25$ to $10e-24$ Nm² as the motor speed increases. They then give a nice summary in the discussion of the possible mechanisms for the stiffening, suggesting that it is caused by a torsion-induced global restructuring of the hook affecting its mechanical properties.

This is a well-written paper that addresses, and answers, an interesting question. The only small comment I have is that it is not clear how the torque shown in figure 5 is obtained from the data. It is explained in the SI, but it would be helpful to include this in the main text. My recommendation is that this paper is accepted for publication in Nature Communications.

Reviewer #3 (Remarks to the Author):

This paper describes the evidence for dynamic torsional strain induced stiffening of the hook, which works as a universal join of the bacterial flagellar motor. The authors used in-line holographic microscopy to track a bead attached to the motor in three dimensions and applied a novel fluctuation analysis to its position data. The authors quantitatively observed interesting features of the tubular biopolymers hook that had been missed by the previous motor rotation analysis by a bead assay. Overall, the paper is reasonably well-written and gives clear conclusion.

Nevertheless, I have a few concerns:

1. While the authors have carefully tried to correct the wall effect on the drag force of the beads, they have treated the underlying assumptions quite simply or crudely. If the beads are relatively small

compared to the cell radius, one might assume that the surface of *E. coli* is a simple plane. On the other hand, if the beads are comparable to or larger than the cell radius, it would be difficult to assume that they are planar rather than cylindrical. Is it reasonable to consider the surface of *E. coli* cells as planar, especially when the size of the beads is large? This assumption is likely to have a significant impact on the values of drag coefficient. As a result, I think that the authors may have overestimated the drag force. I suggest that the validity of the assumption and the error range of the data should be discussed more carefully.

2. As far as I read, the authors ignored the contribution of the flagellar filament stub to estimate the drag ($\gamma\text{-}\phi$) to the motor, but I don't think that's reasonable based on Inoue et al. in 2008, especially when one measures the motor speed beyond the knee speed. I suggest that the drag for a bead assay should be corrected when the beads are small ($R_b = 250 \text{ nm}$).

Minor points

line 49: The first appearance of "*E. coli*" should be written as "*Escherichia coli*".

line 197: "*CheRB*" would be (*italic*)cheRB

line 215: "(i.e. $\langle\theta\rangle$ decreases)" may be "(i.e. $\sigma\text{-}\theta$ and $\langle\theta\rangle$ decreases)"?

line 259: "*Escherichia coli*" could be "*E. coli*".

References: In some references, the author's first name is written with initials, while in others it is written with full spelling. Please check carefully.

Figure2 in SI: I was wondering if the 3D trajectory of the fourth line of Figure 2 shows the opposite phenomenon from the other data. In other word, as the motor speed increases, the center of the bead moves in the z-axis direction and the radius of rotation decreases. Nevertheless, the EI data vs speed in the right column shows the same tendency as others. It is very strange to me. Please explain why this thing happened.

Reviewer #4 (Remarks to the Author):

In this manuscript, rotating bead assays for microbeads attached to the hook and flagellar stub of non-switching *E. coli* cells are reported. The beads rotate in a circular trajectory. By studying the time-dependent rotation of beads attached to bacteria whose motors are stopped and then their rotation resurrected, the radius of the trajectory was observed to increase as rotation speed increased. At the same time, at higher rotation speeds, less fluctuation in the radius was observed. Based on a simple geometrical model in which radius is dependent on the bending of the hook, fluctuations in the radius are associated with the bending stiffness. In addition, the torque and hence torsional winding of the shooK can be associated with the hydrodynamic resistance and rotation speed. As a result, the authors

can characterize the bending stiffness of the hook as a function of its twist, quantitatively measuring a 10-20 fold stiffening in bending as the torsional strain increases.

There are a few important concerns I have that must be clarified before I would be comfortable believing the accuracy of the analysis.

1. The authors do not fully explain the origin of the observed increase in radius r as rotation rate increases. This is important because if we don't understand better why r increases, can we be confident that the fluctuations in r can be solely attributed to hook bending stiffness?

1a. It is confusing why r is larger for larger beads (Fig 1b). Shouldn't larger beads have slower rotation rates, hence smaller trajectory radii?

1b. Is the change in r as rotation speed increases reversible? This would help understand some mechanisms.

1c. Specifically, I worry that if radius is linked to rotation speed, fluctuations in motor torque and rotation speed might be an additional source of fluctuations in angle θ , which is not taken into account in the current analysis which assumes that fluctuations in θ arise due to thermal motion only. This has the potential to change the reported results. Especially, the fluctuations in rotation rates seem to have similar magnitudes at different speeds, meaning that at low rotation rates the relative fluctuation in motor speed is much larger than at high rotation rates. Could this lead to the observed change in fluctuations?

2. When estimating the torque as a function of rotation rate (lines 173-175), the rotational hydrodynamic resistance (not just translational resistance) is not taken into account? Does that make an appreciable contribution to torque, especially for larger beads?

3. The description of literature can be improved to add important context.

3a. The papers by Block et al (refs 34,35) do provide quantitative evidence for the dynamic stiffening of the torsional compliance of the hook, so the statement on line 46-47 is not quite accurate. I agree

that measurements of the response of bending stiffness are lacking. In the next paragraph this is specified better in the first sentence, but somewhat muddled again in later sentences in a way that perhaps oversells the work as the first measurement of torsion-induced stiffening. Similarly, the last paragraph (lines 251-253) also might be clarified. To be clear, I do think that the measurement of bending stiffness is important on its own, so this is just a request for precision.

3b. The manuscript mentions that variable bending stiffness may allow bundle formation then propulsion, but it is not quite clear how that makes sense. In monotrichous bacteria, the increase in bending stiffness allows bacteria to resume straight swimming after reorientation in the scenario mentioned in lines 40-41. Here, the work of Jabbarzadeh and Fu (*Physical Review E* 97:012402, 2019), as well as Park, Kim, and Lim (*Journal of Fluid Mechanics* 859:586, 2019), have provided additional quantitative models for when compression is able to buckle the relaxed hook and lead to reorientation, showing how variability in the bending stiffness (as measured here) of the hook is important. However, in peritrichous bacteria, the increased stiffness would tend to destabilize bundles and straight swimming after the soft hook allows bundle formation.

Dynamic stiffening of the flagellar hook
Manuscript NCOMMS-21-28189A-Z
Response to the reviewers

We thank the editor and the four reviewers for their comments. Addressing their points of concern has allowed us to substantially improve the quality of the manuscript. We include below a point-by-point response to each reviewer, wherein quotes from the reviewers are given in *blue*. We also include details of a separate minor correction that we made during revision. We include revised copies of the manuscript and SI, where changes are shown in blue. We have also updated the online code¹ to reflect the changes. Thank you again for your time and consideration.

On behalf of all the authors,
Sincerely yours,
Francesco Pedaci

Correction of the drag expression

We have corrected the formula of γ_{tg} given in SI sec.2.1 (from $\gamma_{tg}^2 = \gamma_{\parallel}^2 \sin^2 \theta + \gamma_{\perp}^2 \cos^2 \theta$ to $\gamma_{tg} = \gamma_{\parallel} \sin^2 \theta + \gamma_{\perp} \cos^2 \theta$), which now correctly combines the two components of the drag, as explained in the new footnote of the same paragraph. This has the consequence to change the expression of γ_{θ} in eq.1 of the main text and SI eq.1, now corrected, which enters into the algorithm that quantifies EI , with effects propagating in fig.4 and 5 of the main text, and fig.3 of the SI. These figures have been updated, and the resulting changes are very minor in all plots. The message of the updated manuscript remains qualitatively and quantitatively intact. A typo in SI eq.6 has been corrected as well.

Reviewer 1

[...] Here, Nord et al. tracked filament-tethered micro beads to quantify hook bending stiffness of the E. coli flagellum in order to investigate the hook's dynamic mechanical response under changing conditions. They found that the hook functions as a strain-stiffening polymer, which shows a linear increase in hook-bending stiffness as a function of motor torque. This is a nicely written manuscript with beautifully illustrated figures that greatly enhances our mechanistic understanding of one of nature's most fascinating molecular machines, the flagellum. I only have a few comments that might help the authors to improve the clarity of their manuscript:

We thank the reviewer for these positive comments.

¹https://github.com/SMADynamics/BFM_radial_fluct

1) *The number of analyzed motors should be indicated in the figure legends, as well as if the presented data are from multiple motors of a single cell or from motors of different cells.*

We have added to the caption of fig.4 and 5 the text :

The measurements, each on a different cell, consist of 36, 35, and 7 traces for motors at steady state and 6, 13, and 6 resurrection traces for beads of radius $R_b = 1000, 500, 250$ nm, respectively.

2) *The legends of Fig. 2 and Fig. S4 are missing/cut short?*

In our PDF version the legends are correct. We are sorry for the inconvenience.

3) *As stated above, the length of the hook structure is rather tightly controlled (e.g. 55 nm +/- 6 nm in Salmonella (Hirano, T., Yamaguchi, S., Oosawa, K., and Aizawa, S. (1994). Roles of FliK and FlhB in determination of flagellar hook length in Salmonella typhimurium. J Bacteriol 176, 5439-5449)? It is still unclear, why bacteria evolved such an intricate length control mechanism of the hook. The discussion could benefit from discussing the importance of the length control for the hook bending stiffness.*

We agree with the reviewer, and we have added to the Discussion the sentence:

The tight regulation of the physical length of the hook observed in bacteria [8, 20] is likely in line with the need for careful tuning of its elastic properties.

4) *Why does the fliC::Tn10 mutation result in a “sticky” filament?*

This is an error of notation on our part. This has been corrected to read

The *Escherchia coli* strain used was MT03 (parent strain: RP437, $\Delta pilA$, $\Delta cheY$, $fliC^{st}$), a non-switching strain in which chromosomal replacement of the wild-type flagellin gene *fliC* with the *fliCst* variant exposes the hydrophobic core of the flagellar filament, rendering it ‘sticky’ to hydrophobic surfaces such as polystyrene (Kuwajima, 1988).

5) *Does the cheRB::cat insertion have polar effects on cheYZ expression? In this respect, the rationale of using a cheRB mutant (resulting in motors that switch between CCW and CW rotational direction) instead of a cheZB mutant (which would allow to analyze CW-only rotating motors) is not clear.*

The $\Delta cheRB$ mutant, deleted for the receptors methyltransferase and methylesterase, contains a higher concentration of CheY-P than wild-type cells, and it has been shown that they spend a larger percentage of time rotating CW than wild-type cells (Lele et al, 2013). The aim with this mutant was to measure both the CCW and CW rotation on the same motor and hook, as opposed to measuring separate populations of CCW-only and CW-only mutants. The construction was designed in such a way as to avoid any polar effect, in particular on the *cheYZ* genes located downstream of the deleted *cheRB* genes. We have changed the text (SI sec.9.1) to clarify.

Reviewer 2

[...] This is a well-written paper that addresses, and answers, an interesting question. The only small comment I have is that it is not clear how the torque shown in figure 5 is obtained from the data. It is explained in the SI, but it would be helpful to include this in the main text. My recommendation is that this paper is accepted for publication in Nature Communications.

We thank the reviewer for the positive comments. A similar doubt was raised by Reviewer 4 (point 2). We have changed the main text about the definition of torque (from line 172 in sec. "Hook bending stiffness" of the main text) into:

Fig.5a shows the individual trajectories of each hook in the plane (τ or twist, EI), where the torque produced in each time-window of the traces is calculated by $\tau = \gamma_\phi \omega$, where ω is the angular speed of the bead and γ_ϕ is the drag, which includes the rotation and translation of the bead as well as as the correction due to the proximity to the membrane (see sec. "Drag coefficients" and Supplementary sec.2).

Reviewer 3

This paper describes the evidence for dynamic torsional strain induced stiffening of the hook, which works as a universal join of the bacterial flagellar motor. The authors used in-line holographic microscopy to track a bead attached to the motor in three dimensions and applied a novel fluctuation analysis to its position data. The authors quantitatively observed interesting features of the tubular bio-polymers hook that had been missed by the previous motor rotation analysis by a bead assay. Overall, the paper is reasonably well-written and gives clear conclusion.

We thank the reviewer for these comments.

Nevertheless, I have a few concerns:

1. While the authors have carefully tried to correct the wall effect on the drag force of the beads, they have treated the underlying assumptions quite simply or

crudely. If the beads are relatively small compared to the cell radius, one might assume that the surface of E. coli is a simple plane. On the other hand, if the beads are comparable to or larger than the cell radius, it would be difficult to assume that they are planar rather than cylindrical. Is it reasonable to consider the surface of E. coli cells as planar, especially when the size of the beads is large? This assumption is likely to have a significant impact on the values of drag coefficient. As a result, I think that the authors may have overestimated the drag force. I suggest that the validity of the assumption and the error range of the data should be discussed more carefully.

We are aware of the crude simplification of the planar geometry for the surface. Most, if not all, papers which use a bead assay to study the BFM choose to ignore the presence of the surface (see, for example, the typical protocol given by Kasai et al, 2017 ²). We believe our simplified correction to be a better starting point, though we agree that it could be improved. However, the complexity facing better approximations is quite high: the cell, a couple microns in length, can be considered flat on the axial direction, and curved (for micron size beads, as pointed by the reviewer) in the perpendicular direction. The position of the motor on the cell membrane, and its distance from the glass surface are also relevant (e.g. the motor can be positioned laterally on the cell), but we have no direct access to it. For these reasons, it is not easy to estimate to what degree our correction under- or over-estimates the actual drag. A better geometrical approximation, based on new data, could be the object of a subsequent work. Crucially, we note that more accurate corrections to the drag (therefore to the torque value) will not change the main result reported here regarding the hook's torsionally induced increase in bending rigidity. In line with the reviewer's comment, we have added the following text to SI sec.2:

Here we aim at writing the corrections to the drag coefficients due to the proximity with surfaces. In this attempt, we simplify the geometry assuming the simplest case of an infinite plane hard wall, with the hook exiting perpendicularly from it. The actual system is clearly more complicated: the elongated cell has a length of a couple microns and a diameter of $\sim 0.5 \mu\text{m}$; therefore, while along the axial direction it could be considered flat (when using micron-size beads), in the perpendicular direction, the surface curvature is comparable with the bead size. Moreover, the actual position of the motor on the cell surface, and consequently the presence and relative position of the flat glass surface, are also relevant, but have been neglected for simplicity and because we do not have direct access to them.

2. As far as I read, the authors ignored the contribution of the flagellar filament stub to estimate the drag (γ -phi) to the motor, but I don't think that's reasonable based on Inoue et al. in 2008, especially when one measures the motor speed beyond the knee speed. I suggest that the drag for a bead assay should be corrected when the beads are small ($R_b = 250 \text{ nm}$).

²Kasai T., Sowa Y. (2017) Measurements of the Rotation of the Flagellar Motor by Bead Assay. In: Minamino T., Namba K. (eds) The Bacterial Flagellum. Methods in Molecular Biology, vol 1593. Humana Press, New York, NY. https://doi.org/10.1007/978-1-4939-6927-2_14

Inoue et al.³ realize that the contribution of the stub is negligible for $R_b \geq 500$ nm. For the smallest beads we employ ($R_b = 250$ nm), the correction to the drag due to the FliC stub would increase the value of the torque for the green points and traces in fig.5 of the main text. The drag of a $R_b = 250$ nm bead, rotating and translating on a circular trajectory of radius 100 nm (as measured in fig.4a of the main text), is $\gamma_\phi = 0.44(0.60)$ pN nm s in bulk (surface-corrected with 5 nm gap). Considering the flagellar stub as an elongated ellipsoid, of 10 nm diameter, rotating around one end, a comparable drag⁴ (0.54 pN nm s, in bulk) would be reached for a stub length of 1.8 μ m. We can't measure directly the length of the stub left after mechanical shear of the flagella, but this seems too long given that the measured radius of the bead trajectory r is narrowly distributed around $\sim 100 \pm 20$ nm for $R_b = 250$ nm (fig.4a of the main text). If the stub length remains below 800 nm, which is reasonable, the correction to the drag of the bead ($R_b = 250$ nm) is below 10%. For these reasons we did not correct the drag for the smallest beads we employed.

Minor points:

line 49: The first appearance of “E. coli” should be written as “Escherichia coli”.

This has been corrected.

line 197: “CheRB” would be (italic)cheRB

This has been corrected.

line 215: “(i.e. decreases)” may be “(i.e. sigma-theta and decreases)”?

This has now been changed to read “(i.e. both σ_θ and $\langle \theta \rangle$ decrease).

line 259: “Escherichia coli” could be “E. coli”.

This has been corrected.

References: In some references, the author’s first name is written with initials, while in others it is written with full spelling. Please check carefully.

This has been corrected.

Figure2 in SI: I was wondering if the 3D trajectory of the fourth line of Figure 2 shows the opposite phenomenon from the other data. In other word, as the motor speed increases, the center of the bead moves in the z-axis direction and the radius of rotation decreases. Nevertheless, the EI data vs speed in the right column shows the same tendency as others. It is very strange to me. Please explain why this thing happened.

³Inoue, Y., Lo, C. J., Fukuoka, H., Takahashi, H., Sowa, Y., Pilizota, T., ..., Ishijima, A. (2008). Torque–speed relationships of Na⁺-driven chimeric flagellar motors in Escherichia coli. *Journal of Molecular Biology*, 376(5), 1251-1259.

⁴Che, Y. S., Nakamura, S., Kojima, S., Kami-ike, N., Namba, K., Minamino, T. (2008). Suppressor analysis of the MotB (D33E) mutation to probe bacterial flagellar motor dynamics coupled with proton translocation. *Journal of bacteriology*, 190(20), 6660-6667.

We agree that this trace departs from the average behavior, and we included in the SI for completeness. Here the quasi-hemispheric trajectory of the bead seems reversed, with the bead starting low and ending higher in z at higher speed. However, as speed increases, the value of EI increases as for the other motors. We have observed 2 more cases where z slightly increases (instead of decreasing), out of 15 traces where we have measured the x, y, z positions of the bead, the one shown here being the most striking. A reversed hemispheric trajectory can appear if the motor is located at the bottom part of the cell, pointing towards the glass surface (this can happen if the cell is lifted slightly from the glass surface, e.g. if another cell is below it). But then the bead should appear to rotate CW instead of CCW (in our ΔCheY mutants), and this particular bead rotates CCW like all others. So we don't have a clear explanation for this inverted geometry. Importantly, here we do not assume L constant, but we rely on the measured x, y, z to obtain both the angle θ (from $\theta = \arctan(z/\sqrt{x^2 + y^2})$, see point 4.c in SI sec.3) and the theoretical drag γ_θ (point 7.i SI sec.3). From θ we obtain EI , and from γ_θ (comparing with the theoretical value) we estimate the gap between bead and surface, which influences the value of the motor torque. Therefore the plot of EI versus torque is robust against this different geometry. We have added the following text to the caption of SI fig.2:

The second to last line shows one x, y, z trajectory with an inverted convex geometry. Like the others, the bead rotates in the CCW direction, and EI increases with speed. This is a rare case, for which we do not have a clear explanation.

Reviewer 4

In this manuscript, rotating bead assays for microbeads attached to the hook and flagellar stub of non-switching E. coli cells are reported. The beads rotate in a circular trajectory. By studying the time-dependent rotation of beads attached to bacteria whose motors are stopped and then their rotation resurrected, the radius of the trajectory was observed to increase as rotation speed increased. At the same time, at higher rotation speeds, less fluctuation in the radius was observed. Based on a simple geometrical model in which radius is dependent on the bending of the hook, fluctuations in the radius are associated with the bending stiffness. In addition, the torque and hence torsional winding of the hook can be associated with the hydrodynamic resistance and rotation speed. As a result, the authors can characterize the bending stiffness of the hook as a function of its twist, quantitatively measuring a 10-20 fold stiffening in bending as the torsional strain increases. There are a few important concerns I have that must be clarified before I would be comfortable believing the accuracy of the analysis.

We thank the reviewer for their very relevant comments, and address each of the concerns raised below.

1. The authors do not fully explain the origin of the observed increase in radius r as rotation rate increases. This is important because if we don't understand

better why r increases, can we be confident that the fluctuations in r can be solely attributed to hook bending stiffness?

We cannot clearly identify a single mechanism that explains the observed increase in radius, and the consequent decrease in angle θ . In the discussion and in SI sec.6, we put forward two possible hypotheses. The first possibility is that the bead is pushed towards the membrane by the motor torque, which is redirected by the bent hook. Due to the motor rotation and bending of the hook, the bead would tend to rotate around the axis of the FliC stub (which, accommodating the large bead, cannot be perfectly vertical), until it is stopped by the interaction with the membrane. A second possible mechanism, discussed in SI sec.6, is that the complex structure of the hook could intrinsically couple twist and bend, as was found in other biopolymers like DNA and actin filaments. We hope that future single molecule torque spectroscopy measurements on purified hooks can one day test these hypotheses.

1a. it is confusing why r is larger for larger beads (Fig 1b). Shouldn't larger beads have slower rotation rates, hence smaller trajectory radii?

It is true that, for a given bead, slower rotation gives a smaller radius, r . It is also true that, because of the motor mechanism, larger beads are rotated more slowly at steady state than smaller beads. However, the x, y position we measure is that of the center of the bead; for this reason alone, beads of larger radius R_b will give larger r . This is complicated by the fact that the microscopic geometry of the tether, including the allowed bending angle of the hook, is constrained by the size of the bead, but in general larger r values with larger beads is explained by the fact that to geometrically define the center of the measured trajectory one has to include the bead radius.

1b. Is the change in r as rotation speed increases reversible? This would help understand some mechanisms.

While we did not explicitly test this hypothesis, we have indirect evidence that the change in r with speed is reversible. During experiments, all of the bacteria in the flow cell are exposed to the ionophore multiple times, though only a few can be recorded at once. Therefore, the resurrection of a motor, and the described characteristic behavior of radius, angle, and speed, can be observed after several (up to 5) rounds of Proton Motive Force depletion, which imposes zero motor speed each time it's inserted in the flow cell. This suggests that the bead follows a similar trajectory in r and θ each time it starts rotating from rest.

1c. Specifically, I worry that if radius is linked to rotation speed, fluctuations in motor torque and rotation speed might be an additional source of fluctuations in angle theta, which is not taken into account in the current analysis which assumes that fluctuations in theta arise due to thermal motion only. This has the potential to change the reported results. Especially, the fluctuations in rotation rates seem to have similar magnitudes at different speeds, meaning that at low rotation rates the relative fluctuation in motor speed is much larger than at high rotation rates. Could this lead to the observed change in fluctuations?

We describe the fact that in general θ tends to decrease with increasing torque (or speed). Due to this relationship, the referee is correctly concerned with the possibility that fluctuations in motor torque (related to the stochastic dynamics of the motor) could contribute to the measured fluctuations in θ , therefore affecting the value of EI (proportional to $1/\sigma_\theta^2$, where σ_θ^2 is the variance of θ in a time window), the true value of which should rely on thermal fluctuations only. For the reasons below, we believe that the proposed twist-dependent increase in bending stiffness of the hook polymer is the best explanation of our data.

1. As stated, θ is observed to decrease with speed or torque. The plot of θ versus torque for all the traces (see fig.1a of this document) shows that overall, for the three measured bead sizes and with a large variability among traces, the decrease in θ with torque reaches a plateau at a torque of ~ 500 pN nm. On the other hand, the plot of EI versus torque (see fig.1b of this document, same as fig.5a of the main text) increases up to torques of ~ 1000 pN nm. This difference in “cut-off torque” value between EI and θ as a function of torque suggests that the change in EI with torque is not mainly driven by the change in θ . So we conclude that, if present, this spurious effect would affect the values of EI mainly at torque below 500 pN nm, and not above. The low torque regime is addressed by the next point.
2. Both the absolute values of EI and its trend with increasing torque, which we obtain from the fluctuations of θ , are in agreement with the results described by Son et al.⁵ (see also Table 1 in our SI), where the relaxed (zero torque) and loaded (high torque) hook are considered. In particular, the value of EI of the relaxed hook (found lower than the one measured at physiological swimming speed) was measured by Son et al. on motors where the speed was truly zero, observing fluctuations of pure thermal origin. In our work, instead, a small positive torque is required for the measurement. The two works, with different techniques, yield comparable values of EI for the relaxed state (for us at torque lower than 500 pN nm), the state in which the variation of θ with torque (see point 1 above) and torque fluctuations are the highest, and therefore where the artifact described by the reviewer should be most visible. This builds further confidence that our measurements are free from such an artifact, also at low torque.
3. If speed fluctuations (or relative speed fluctuations) affect the value of EI as pointed out by the reviewer, a trend should be apparent in the plot of EI versus σ_ω (or σ_ω/ω) encompassing the three loads measured (where σ_ω is the standard deviation of the angular speed ω in a time-window). In fig.1c,d of this document, such a clear relationship cannot be observed for the three loads in the plots of EI versus speed or relative speed. On the contrary, we find a clear relation among the three loads only in the plot of EI versus torque, which supports our conclusion.

A discussion of the possible effect of torque fluctuations on the measurement of EI , based on the above points, has been added to the SI sec. “Effect of torque fluctuations”.

2. When estimating the torque as a function of rotation rate (lines 173-175), the rotational hydrodynamic resistance (not just translational resistance) is not taken

⁵Son, Kwangmin, Jeffrey S. Guasto, and Roman Stocker. “Bacteria can exploit a flagellar buckling instability to change direction.” Nature physics 9.8 (2013): 494-498.

Figure 1: a) Change of angle θ , averaged in each time-window of all the traces. b) Hook bending stiffness EI as a function of torque (same as fig.5a of the main text). c) Hook bending stiffness EI as a function of angular speed standard deviation σ_ω . d) Hook bending stiffness EI as a function of relative speed standard deviation σ_ω/ω . As in the main text, the colors green, orange and blue correspond to beads of radius $R_b = 250, 500, 1000$ nm respectively.

into account? Does that make an appreciable contribution to torque, especially for larger beads?

The rotational drag is taken into account. In SI sec. 2.2 we define the drag coefficient γ_ϕ of the bead moving in a plane parallel to x, y , used to calculate the torque. It includes one component for the rotation of the bead around its axis, and one component for its rotation along a circular trajectory. The expression takes also into account the effect of the wall proximity, in Faxen and Brenner formalism. Please, see the reply to Reviewer 2 for the change in the main text.

3. The description of literature can be improved to add important context.

3a. The papers by Block et al (refs 34,35) do provide quantitative evidence for the dynamic stiffening of the -torsional- compliance of the hook, so the statement on line 46-47 is not quite accurate. I agree that measurements of the response of -bending- stiffness are lacking. In the next paragraph this is specified better in the first sentence, but somewhat muddled again in later sentences in a way that perhaps oversells the work as the first measurement of torsion-induced stiffening. Similarly, the last paragraph (lines 251-253) also might be clarified. To be clear, I do think that the measurement of bending stiffness is important on its own, so this is just a request for precision.

We understand the reviewer's desire for precision on these points. The statement on line 46-47 has been changed to read,

measurements of the hook's dynamic bending rigidity under changing conditions are lacking.

We rephrase the relevant portion in the following paragraph to read,

Earlier measurements have provided evidence for a torsional-strain induced increase in the torsional stiffness [papers by Block et al cited]. Here we provide quantitative, *in-vivo*, and time-resolved evidence for a dynamic torsional-strain induced increase in the bending stiffness of the bacterial hook [...]

Finally in the last paragraph, we have clarified in a couple places that we refer specifically to *bending* stiffening.

3b. The manuscript mentions that variable bending stiffness may allow bundle formation then propulsion, but it is not quite clear how that makes sense. In monotrichous bacteria, the increase in bending stiffness allows bacteria to resume straight swimming after reorientation in the scenario mentioned in lines 40-41. Here, the work of Jabbarzadeh and Fu (Physical Review E 97:012402, 2019), as well as Park, Kim, and Lim (Journal of Fluid Mechanics 859:586, 2019), have provided additional quantitative models for when compression is able to buckle the relaxed hook and lead to reorientation, showing how variability in the bending stiffness (as measured here) of the hook is important. However, in peritrichous bacteria, the increased stiffness would tend to destabilize bundles and straight swimming after the soft hook allows bundle formation.

We agree with the reviewer that this point is not intuitive, and we thank the reviewer to pointing us to these two articles, which are relevant and have now been included in the last paragraph of our discussion. These works assume a straight hook, which is reasonable for *V. alginolyticus*. However, in peritrichous bacteria, the hook is supercoiled and bent in its native state. We think the stiffening can actually stabilize bundle formation, but a complete model of the stiffening hook and bundle mechanics is needed to test this claim. We have rewritten the last paragraph of the discussion to clarify this point.

While the hook of *V. alginolyticus* appears straight [Hosogi 2011, Koike 2010], that of *S. enterica* is supercoiled [Kato 2019]. Thus, the angle between the hook and membrane is natively acute, and our experiments suggest that this angle decreases slightly with twist. One may imagine that an increasingly rigid hook can produce opposite effects on the stability of the flagellar bundle in peritrichous bacteria. On one hand, a more rigid hook can withstand the bending moment coming from a tilted flagellum rotating in the bundle, increasing bundling stability. On the other hand, the universal joint function of the hook could conceivably be negatively affected by increasing bending rigidity. Our results indicate that the stabilizing effects prevail, constraining the mechanical model of a stiffening hook.

REVIEWER COMMENTS

Reviewer #1 (Remarks to the Author):

Nord et al. substantially revised their manuscript and appropriately addressed my concerns. I recommend publication of the manuscript.

Reviewer #3 (Remarks to the Author):

The authors have adequately addressed all concerns.

I recommend that this paper be published in Nature communications.

Reviewer #4 (Remarks to the Author):

In the reply and revised manuscript, the authors have addressed most of my comments and concerns (original reviewer #4), except for my first, about whether motor fluctuations could contribute significantly to observed fluctuations in theta, introducing errors in the value of EI. I would only support publication if this concern can be addressed. Without clearing this up, it is possible that their main conclusions are an artifact of this confounding effect.

In their response, to this, point 2, the authors point out that their value of EI for the relaxed hook ($1.2 \times 10^{-25} \text{ N m}^2$) is comparable to that of Son et al (for *V. alginolyticus* ($3.6 \times 10^{-26} \text{ N m}^2$)). In fact, their value is larger, but additional fluctuations in theta arising from motor rotation fluctuations should instead tend to decrease EI. So I think this is reasonably good evidence (albeit with a difference in species).

However, I find their response in point 3 a little lacking. They show that there is no trend in EI as a function of σ_{ω} , from which I conclude that σ_{ω} would not be the sole determinant of fluctuations that contribute to their estimate of EI, not that it has no contribution. I think there is no

question that there must also be thermal fluctuations even if fluctuations in rotation rate have some contribution.

Upon reflection, what I think would be a convincing test about whether the motor fluctuations are an important contribution would be to use the derived relationship between θ and ω [$\theta(\omega)$, e.g., Fig 3(c2)] to estimate how much variation in θ (σ_{expected}) would be expected for the observed fluctuation in ω at different ω s or θ s. Then, to plot the relative fraction of observed σ_{θ} arising from the motor torque ($\sigma_{\text{expected}}/\sigma_{\theta}$) as a function of torque or motor speed. This would clearly show whether the motor fluctuation effect is large relative to observed variation of θ or not, for different regimes of torque.

Dynamic stiffening of the flagellar hook
Manuscript NCOMMS-21-28189B
Response to the reviewers

We thank again the editor and the reviewers for their positive comments. Here we respond to the points raised by reviewer 4.

Reviewer 4 (Remarks to the Author):

“ In the reply and revised manuscript, the authors have addressed most of my comments and concerns (original reviewer 4), except for my first, about whether motor fluctuations could contribute significantly to observed fluctuations in theta, introducing errors in the value of EI. I would only support publication if this concern can be addressed. Without clearing this up, it is possible that their main conclusions are an artifact of this confounding effect.

In their response, to this, point 2, the authors point out that their value of EI for the relaxed hook ($1.2 \times 10^{-25} Nm^2$) is comparable to that of Son et al (for *V. alginolyticus* ($3.6 \times 10^{-26} Nm^2$)). In fact, their value is larger, but additional fluctuations in theta arising from motor rotation fluctuations should instead tend to decrease EI. So I think this is reasonably good evidence (albeit with a difference in species).

However, I find their response in point 3 a little lacking. They show that there is no trend in EI as a function of `sigma_omega`, from which I conclude that `sigma_omega` would not be the sole determinant of fluctuations that contribute to their estimate of EI, not that it has no contribution. I think there is no question that there must also be thermal fluctuations even if fluctuations in rotation rate have some contribution.

Upon reflection, what I think would be convincing test about whether the motor fluctuations are an important contribution would be to use the derived relationship between theta and omega [`theta(omega)`, e.g., Fig 3(c2)] to estimate how much variation in theta (`sigma_expected`) would be expected for the observed fluctuation in omega at different omegas or thetas. Then, to plot the relative fraction of observed `sigma_theta` arising from the motor torque (`sigma_expected/sigma_theta`) as a function of torque or motor speed. This would clearly show whether the motor fluctuation effect is large relative to observed variation of theta or not, for different regimes of torque. ”

Response

Following the mechanisms proposed by the reviewer, we have run further analysis to quantitatively weight the role of speed fluctuations in bending. In doing so, we have performed the test suggested by the reviewer, also expanding it including a reasonable filtering of the speed trace, as described below. We find that the implementation of the proposed test is not straightforward, as it depends on several details that we address below. As we show below, we generally see a great heterogeneity of the results. While the mechanism proposed by the reviewer is *a priori* reasonable, we cannot find a general quantitative (nor qualitative) trend

in the analysis. Due to these difficulties, we conclude that the mechanism is not supported by this kind of test, and that the interpretation we give of dynamic stiffening of the hook remains valid to explain our data. In the main text we have added the following text:

In Supplementary sec.7 we consider the effects of speed fluctuations combined with the observed decrease of $\langle\theta\rangle$ with speed, concluding that the effect of speed fluctuations provide a large variability of results, and can not explain the observed trend of EI increasing with speed.

The derived relationship between θ and ω

To define this relationship, we first took a global average over all the traces (for a same load), and we found that, once averaged, θ does not show a substantial change with ω , because the spread between different traces washes out details that are apparent only on individual traces. Therefore we consider one resurrection trace at the time, and we fit each θ Vs ω with an ad-hoc function (polynomial or exponential). We see that $\theta(\omega)$ clearly decreases in 12/17 traces for 1 micron beads, and 2/6 traces for the 2 micron beads. However the data do not clearly suggest a unique type of dependency (eg only exponential). Not having a mechanistic model describing this relationship, the choice of the fit function remains arbitrary, and we see a high degree of variability in the goodness of such fits.

Speed fluctuations σ_ω

Importantly, σ_ω is always large with respect to ω . This is because $\omega(t)$ is the time-derivative of the noisy signal $\phi(t)$ (the derivative makes small position fluctuations become large speed noise). In particular, negative values of ω are very frequent. To qualitatively show readable speed traces, when acquiring at high sampling frequency, it is always necessary to filter the high frequency content of $\omega(t)$ (as in fig.3a of the main text, and in virtually all BFM studies with fast acquisition). Following and expanding the mechanism proposed by the reviewer, below we will use both the unfiltered and one filtered version of $\omega(t)$, where the latter effectively reduces σ_ω and therefore $\sigma_{\theta,e}$ (where we indicate `sigma_expected` with $\sigma_{\theta,e}$). In the spirit of the proposed mechanism, we can add the hypothesis that speed fluctuations are filtered by the elastic bending of the hook before producing bending fluctuations. Therefore the filtered version of the speed trace used below will be obtained by low-pass filtering $\omega(t)$ at a cut-off frequency equal to the corner frequency of the Lorentzian that successfully fits the power spectral density of $\theta(t)$, in each time-window of the trace (as in fig.3d1 of the main text).

Quantifying $\sigma_{\theta,e}$

Once the fit of θ Vs ω is defined for a single trace, $\sigma_{\theta,e}$ corresponds, at each speed ω , to the interval of θ within $\omega \pm \sigma_\omega$. Because σ_ω (especially for the unfiltered trace) is often large, defining $\sigma_{\theta,e}$ depends on whether we extend the fit beyond the experimental values visited by ω . We choose to expand the interval of possible values of ω by 10-50% beyond the maximum and minimum experimental ω values. With all the assumptions and choices described above, we can finally quantify $\sigma_{\theta,e}$ and the corresponding EI_e (the expected bending stiffness), comparing them to the experimental σ_θ and corresponding EI directly measured from x, y .

Figures description

In the figures below, each row corresponds to one resurrection trace. The first column shows the experimental values of θ Vs ω (dark points, measured from x, y) and the resulting fit (yellow line). The second column shows the speed fluctuations σ_ω measured from the full (blue points) and from the filtered (red points) speed trace. In the third column, the fluctuations of θ are shown, measured directly from the experimental trace (black), and inferred, according to the mechanism suggested by the reviewer, from the unfiltered (blue points) and filtered (red) σ_ω , via the fit of θ Vs ω . Therefore, $\sigma_{\theta,e}$ is indicated by the blue and red points, and needs to be compared with the observed black points. The fourth column shows the corresponding values of EI , following the same color code. For a same load, we qualitatively sort the figures by goodness of the fit of θ Vs ω .

Comments

Considering the unfiltered speed trace (blue points in columns 2,3,4), the corresponding $\sigma_{\theta,e}$ is found larger (10/23), smaller (9/23) or of the same order as (3/23) the experimental σ_θ (black points). Without filtering the speed trace, σ_ω produces almost always an interval of speeds larger than the allowed range defined in the fit of θ Vs ω , which explains why $\sigma_{\theta,e}$ in the unfiltered case shows almost no speed dependency (blue points, column 3).

Filtering $\omega(t)$ at the bending corner frequency (red points in columns 2,3,4) decreases σ_ω (red points in column 2) and the corresponding $\sigma_{\theta,e}$ (red points in column 3). In some cases, filtering also changes the slope of σ_ω Vs ω , with respect to the unfiltered trace (column 2). With a filtered ω , we find $\sigma_{\theta,e}$ smaller (15/23) or of the same order as (8/23) the experimental σ_θ (black points). In general, and in particular when found of the same order of magnitude as the experimental σ_θ , we note that $\sigma_{\theta,e}$ does not have the same dependency on ω as the experimental σ_θ .

In conclusion, this kind of quantification is not of straightforward implementation, and we have described the different assumptions that are required. The variability of the behaviors and results is large, often with orders of magnitude differences. Without filtering the speed trace, the result of the analysis is widely variable, and would not support the proposed mechanism as a deterministic cause to our observations. Filtering the speed, assuming it is reasonable to do so, produces mainly small $\sigma_{\theta,e}$, or, when of the right order, not with the right speed dependency. The main critical point is likely the high variability in the way θ changes with ω . In the manuscript we avoided to describe it more than as a general decreasing trend, because the results we describe for EI are robust with respect to it. The mechanism proposed by the reviewer is instead very dependent on the details of this relationship. Because of this variability, a precise quantification of the weight of the proposed mechanism (as could be provided by a single percentage value) is not possible, thus we have chosen to show here individual results instead of building a global indicator. We believe our interpretation of dynamic stiffening of the hook remains the simplest and most robust interpretation.

Figure 1: $R_b = 1000$ nm

Figure 2: $R_b = 500$ nm

Figure 3: $R_b = 500$ nm

Figure 4: $R_b = 500$ nm

REVIEWERS' COMMENTS

Reviewer #4 (Remarks to the Author):

I am satisfied with the authors' response showing that the fluctuations in in speed do not adequately explain the observed trends in fluctuations in bending angle theta. In my opinion the manuscript is suitable for publication after the authors make the minor clarifications below:

The authors mention that the speed in Fig 3a is filtered. I did not appreciate this fact from the descriptions in the text and SI. The authors should explain somewhere explicitly how the speed is actually determined. This is particularly important since if the speed has large errors from the noise in phi, then the derived torque would also have similar uncertainty. So knowing how those high frequency fluctuations are removed is essential. Presumably to be consistent, the time window for measuring the speed should be similar to the time window used to measure the fluctuation spectrum of theta.

Dynamic stiffening of the flagellar hook
Manuscript NCOMMS-21-28189B
Response to the reviewers

We thank the editor and reviewer 4 for their positive comments. Here we respond to the point raised by reviewer 4.

Reviewer 4 (Remarks to the Author):

“ I am satisfied with the authors’ response showing that the fluctuations in in speed do not adequately explain the observed trends in fluctuations in bending angle theta. In my opinion the manuscript is suitable for publication after the authors make the minor clarifications below: The authors mention that the speed in Fig 3a is filtered. I did not appreciate this fact from the descriptions in the text and SI. The authors should explain somewhere explicitly how the speed is actually determined. This is particularly important since if the speed has large errors from the noise in phi, then the derived torque would also have similar uncertainty. So knowing how those high frequency fluctuations are removed is essential. Presumably to be consistent, the time window for measuring the speed should be similar to the time window used to measure the fluctuation spectrum of theta. ”

The full procedure to determine the speed is described in (now called) Supplementary Methods 2.2 “Analysis workflow”. The x, y bead trajectory is fit by a circle, found centered in x_0, y_0 . The speed ω is then calculated as the time derivative of the angle $\phi = \arctan((y - y_0)/(x - x_0))$. When plotting ω (and only when plotting it) we filter the trace as mentioned in the captions of fig.1c and fig.3a: (“to increase readability the trace is here filtered with a 8 ms running window median filter”). The value of the torque (shown in fig.5) is calculated from the unfiltered speed trace, averaged in the same time-windows along the trace that are used for our fluctuation spectral analysis (one average torque value per time-window). Therefore the torque is unaffected by the filtering of the speed, which is only performed to increase readability of the traces in the figures.

For more clarity we have modified the “Data analysis” section of the main text as follows: “In each window, the angle $\theta(t)$ (see Fig.2) was calculated from the $(x(t), y(t))$ or $(x(t), y(t), z(t))$ position of the bead by fitting the bead trajectory to an ellipse (see Supplementary Methods 2.2).”